

# Using high-resolution melting to identify Calliphoridae (blowflies) species from Brazil

Pablo Viana Oliveira[1], Francine Alves Nogueira de Almeida[1], Magda Delorence Lugon[1], Karolinni Bianchi Britto[1], Janyra Oliveira-Costa[2], Alexandre Rosa Santos[3] and Greiciane Gaburro Paneto[1,4]

[1] Programa de Pós-Graduação em Biotecnologia, Universidade Federal do Espírito Santo, Vitória, Espírito Santo, Brazil
[2] Polícia Civil do Estado do Rio de Janeiro, Instituto Médico Legal Afrânio Peixoto, Rio de Janeiro, Rio de Janeiro, Brazil
[3] Centro de Ciências Agrárias e Engenharias, Universidade Federal do Espírito Santo, Alegre, Espírito Santo, Brazil
[4] Centro de Ciências Exatas, Naturais e da Saúde, Universidade Federal do Espírito Santo, Alegre, Espírito Santo, Brazil

Corresponding author
Pablo Viana Oliveira,
pablo.viana27@gmail.com

## ABSTRACT

Forensic entomology is the study of insects and other arthropods used in the solution of crimes. Most of entomological evidences strongly depend on accurate species identification. Therefore, new methods are being developed due to difficulties in morphological identification, including molecular methods such as High-Resolution Melting. In this study, we reported a new HRM primer set to identify forensically important Calliphoridae (blowflies) from Brazil. For such purpose, Calliphoridae species of forensic importance in Brazil were listed and confirmed by specialists. Mitochondrial COI sequences of those species were downloaded from databases and aligned, and polymorphic variations were selected for distinction between species. Based on it, HRM primers were designed. Forty-three fly samples representing six species were tested in the HRM assay. All samples had the COI gene sequenced to validate the result. Identifying and differentiating the six species proposed using a combination of two amplicons was possible. The protocol was effective even for old insect specimens, collected and preserved dried for more than ten years, unlike the DNA sequencing technique that failed for those samples. The HRM technique proved to be an alternative tool to DNA sequencing, with advantage of amplifying degraded samples and being fast and cheaper than the sequencing technique.

## INTRODUCTION

Morphology-based species identification, although widely used, may be difficult in cases of closely related species and damaged or degraded specimens (*Galan, Pagès & Cosson, 2012*; *Sontigun et al., 2018b*), and due to the lack of taxonomic keys and specialists (*Chen, Hung & Shiao, 2004*; *Hebert, Ratnasingham & DeWaard, 2003*). Technologies, such as

scanning electron microscopy (SEM), can help giving detailed information on the external morphological characteristics of the body and genitalia of adult blowflies and can help identify the immature forms (*De Oliveira David, Rocha & Caetano, 2008*; *Mendonça et al., 2014a*, *2014b*; *Sontigun et al., 2018a*). The destruction of these characteristics, however, may lead to incorrect identification or non-identification (*GilArriortua et al., 2014*).

DNA barcoding, a powerful molecular tool, was created to overcome these barriers. Based on one or more standardized DNA regions—the mitochondrial Cytochrome Oxidase I gene (COI) in the case of animals (*Hebert et al., 2003*), and chloroplastidial and nuclear regions for plants (*CBOL Plant Working Group, 2009*; *Kress et al., 2005*) and fungi (*Schoch et al., 2012*)—barcoding allows identifying the species with enough accuracy and is applicable to many areas such as tracking of adulterations in food (*Dawnay et al., 2016*), beverages (*De Castro et al., 2017*), medicinal herbs (*Bansal et al., 2018*), identification of flies in forensic area (*Jang et al., 2019*; *Oliveira et al., 2017*; *Rolo et al., 2013*) and species that cannot be determined morphologically (*Klippel et al., 2015*), among others.

Insect specimens are used to estimate the post-mortem interval (PMI) in forensic entomology, which comprises the time from death to the discovery of the human corpse. It is based on the life cycle stage and the succession patterns of insects (*Amendt, Krettek & Zehner, 2004*; *Catts & Goff, 1992*). Furthermore, these insects are used to determine the cause of the death or whether the body was transported to a place other than that of death (*Amendt et al., 2011*; *Zajac et al., 2016*). The blowflies of family Calliphoridae (Insecta, Diptera) colonize the corpse before the other insects and provide most of the information about the PMI (*Cooke et al., 2019*). Families such as Muscidae, Sarcophagidae, Fanniidae and Phoridae are also forensically important, besides other insects of orders Coleoptera, the beetles, Hymenoptera, the wasps, ants and bees, and Lepidoptera, the butterflies (*Amendt, Krettek & Zehner, 2004*).

Usually, DNA barcoding technique uses fresh or preserved tissue samples to extract DNA since long amplicons are necessary (658 bp for COI, for example). However, in many circumstances, recovering sequences is not possible because only degraded DNA is available (*Boyer et al., 2012*), which is the major problem of specimens older than a decade (*Hajibabaei et al., 2006*). High resolution melting (HRM) analysis is an alternative method to sequencing to overcome this problem (*Fernandes et al., 2017*; *Osathanunkul, Osathanunkul & Madesis, 2018*). This is a RT-PCR-based and non-contaminated post-PCR technique that allows analyzing genetic variation in small PCR amplicons (usually 80–120 bp) by detecting small differences in the melting temperatures of the sequences through the melting curves (*Wittwer, 2009*; *Wittwer et al., 2003*). Furthermore, this method is cheaper and faster than Sanger sequencing, being a good alternative to be used by many laboratories (*Wittwer, 2009*). However, the scarcity or unavailability of HRM instruments can be a limitation of the technique. Although routinely qPCR instruments are designed to monitor fluorescence during DNA melting, not all allow performing the HRM analysis (*Li et al., 2012*). Also, the designed primers needs to follow some requirements to be successful, for example, amplification of short fragments, what can limit or hinder the application of HRM in some cases. Furthermore, some

**Table 1 Primer sequences used in this study.**

| Name | Direction | Sequence (5′→3′) | Ta (°C) | Amplicon size (bp) | Source |
|---|---|---|---|---|---|
| HRM_82F | Forward | AGTAGAAAATGGGGCTGGAA | 54 | 82 | This study |
| HRM_82R | Reverse | ATCAACTGATGCTCCTCCAT | | | |
| HRM_124F | Forward | AATGTAATTGTAACAGCTCACG | 56 | 124 | This study |
| HRM_124R | Reverse | GTGGGAAAGCTATATCTGGAG | | | |
| LCO-1490 | Forward | GGTCAACAAATCATAAAGATATTGG | 51 | 658 | *Folmer et al. (1994)* |
| HCO-1490 | Reverse | TAAACTTCAGGGTGACCAAAAAAT | | | |
| C1-J-2495 | Forward | CAGCTACTTTATGAGCTTTAGG | 51 | 304 | *Wells & Sperling (2001)* |
| C1-N-2800 | Reverse | CATTTCAAGCTGTGTAAGCATC | | | |

polymorphisms may not be distinguished by HRM, as there is a better differentiation between C/T and G/A or C/A and G/T than C/G or A/T (*Roche Diagnostics GmbH, 2012*; *Słomka et al., 2017*).

HRM is being used to identify a wide range of organisms such as animals (*Klomtong, Phasuk & Duangjinda, 2016*), plants (*Mishra, Shukla & Sundaresan, 2018*; *Sun et al., 2016*), fungi (*Bezdicek et al., 2016*), bacteria (*Iacumin et al., 2015*), and protozoa (*Aghaei et al., 2014*). In the specific case of insects, it was reported to identify forensic flies (*Malewski et al., 2010*) and mosquitos (*Kang & Sim, 2013*). However, the insect species addressed in each study are specific to each region of the planet. Thus, in this study, we reported a new HRM primer set to identify forensically important Calliphoridae (blowflies) in Brazil.

# MATERIALS AND METHODS

## HRM primer design

Initially, Calliphoridae species forensically important in Brazil were listed based on literature and checked by specialists. Mitochondrial COI barcode sequences of those species were downloaded from BOLD Systems (available at www.boldsystems.org) (*Ratnasingham & Hebert, 2007*) (examples are shown in Supplemental File S2) and aligned through ClustalW in Bioedit software (*Hall, 1999*). Two regions containing polymorphic variations were selected for species distinction (Supplemental File S3). HRM primers were designed using Primer3 software (http://bioinfo.ut.ee/primer3/) (*Koressaar & Remm, 2007*), flanking these regions and generating small amplicon sizes, as shown in Table 1. uMelt Melting Curve Predictions Software version 2.0.2 (available at https://www.dna-utah.org/umelt/um.php) was used to predict the melting temperature of each amplicon, enabling identifying each species based on its temperature variation.

## Sample and DNA isolation

For this study, 43 fly samples were used to validate the HRM assay. These samples included six Calliphoridae species (*Chrysomya albiceps* Wiedemann, 1819, *Chrysomya megacephala* Fabricius, 1794, *Chrysomya putoria* Wiedemann, 1830, *Cochliomyia*

*macellaria* Fabricius, 1775, *Lucilia cuprina* Wiedemann, 1830, and *Lucilia eximia* Wiedemann, 1819), collected from different sites in Southeast Brazil, and identified through identification key (*De Carvalho & De Mello-Patiu, 2008*) and by the specialists Dr. Janyra Oliveira-Costa (Rio de Janeiro Scientific Police) and Dr. Patrícia Jacqueline Thyssen (Unicamp/São Paulo), as shown in the Supplemental File S1. Part of the samples had, approximately, eleven years of collection and was sent to us in dry conditions, another part had 6 years and was preserved in alcohol 70 °G.L and five fresh specimens of *L. cuprina* were supplied in absolute alcohol. Genomic DNA was isolated from 20 mg of thoracic muscles using NucleoSpin® Tissue Kit (Macherey-Nagel, Hoerdt, Germany) according to the manufacturer's instructions. DNA concentration and purity were evaluated using a NanoDrop™ 2000 Spectrophotometer (Thermo Scientific, Waltham, MA, USA). DNA samples were stored at −30 °C for further use.

## PCR-HRM assay and data analysis

HRM with pre-amplification were performed on a LightCycler®96 real-time PCR instrument (Roche Diagnostics, Risch-Rotkreuz, Switzerland). The HRM-PCR reaction mixture (10 µL) contained 5 ng of genomic DNA, five µL of SsoFast™ EvaGreen® Supermix, 500 nM of each primer and ultrapure water up to the final volume. The samples were run in triplicate and a negative control was used in each experiment to exclude contamination. DNA amplification was achieved under the following conditions: first denaturation at 95 °C for 2 min, then 40 cycles of 95 °C for 30 s, annealing at 54 and 56 °C, for 82 and 124 bp amplicon, respectively, for 30 s and extension at 72 °C for 30 s. After amplification, PCR products were denatured at 95 °C for 1 min and renatured at 40 °C to form DNA duplexes. Melting curves were acquired by heating from 65 to 97 °C with 25 data acquisitions per degree. Data were analyzed using LightCycler®96 SW 1.1 version (Roche Diagnostics, Risch-Rotkreuz, Switzerland). Genotypes were identified by examining normalized melting curves, difference and derivative plots of the melting data. Melting temperature (Tm) data were statistically analyzed using Microsoft Excel 2010 and R statistical computing software (*R Development Core Team, 2018*) to the calculation of mean, standard deviation and confidence interval for each sample run in triplicate (Supplemental File S4).

## DNA sequencing

Samples were previously sequenced to validate the results of HRM analysis. For this purpose, all samples were subjected to conventional PCR using universal COI primers. Table 1 shows an alternative primer pair to produce short fragment (385 bp) was used when universal primers failed. PCR products were purified using ExoSAP protocol. Sequencing was performed on an ABI 3500 DNA Sequencer (Thermo Fischer Scientific, USA) using BigDye Terminator Cycle Sequencing Kit version 3.1 (Thermo Fischer Scientific, Waltham, MA, USA) following the manufacturer's instructions. Sequences obtained were confronted with BOLD Systems (http://www.boldsystems.org) and GenBank to identify species. Similarity above 99% was considered identified.

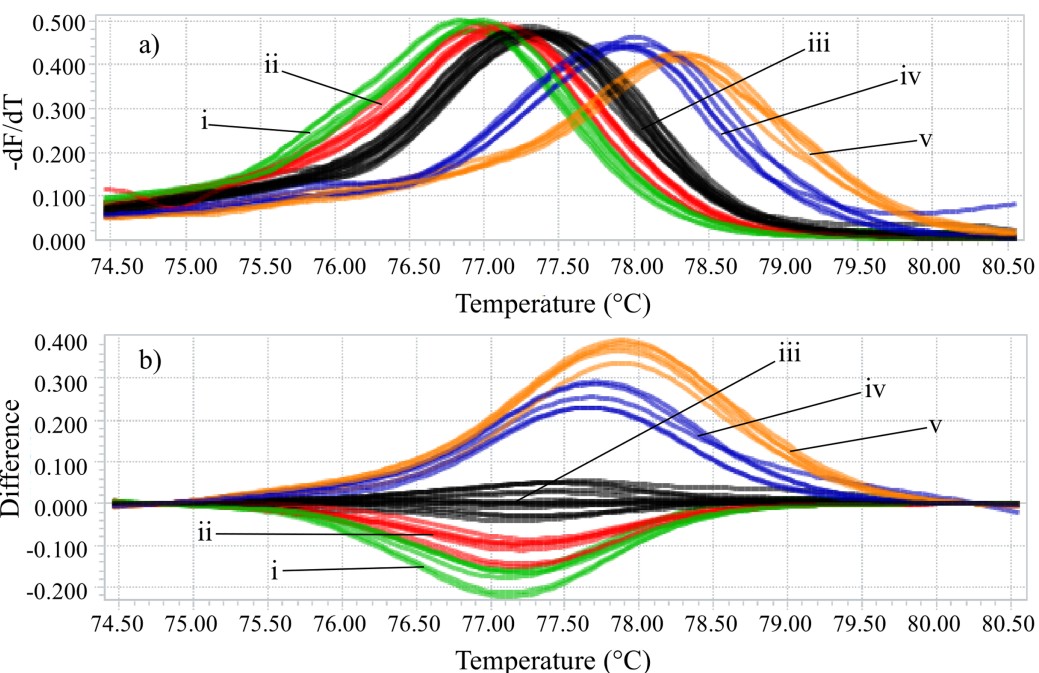

**Figure 1 High-resolution melting analysis using COI primers (82 bp amplicon) for identification of Calliphoridae (blowflies) species.** (A) Normalized melting curves. (B) Difference plot curves using *C. megacephala* as reference genotype. Orange: *L. eximia;* Blue: *L. cuprina;* Black: *C. megacephala* and *C. albiceps*; Red: *Co. macellaria*; Green: *C. putoria*. (i) *C. putoria*; (ii) *Co. macellaria*; (iii) *C. megacephala* and *C. albiceps*; (iv) *L. cuprina*; (v) *L. eximia*.

## RESULTS AND DISCUSSION

Identifying the six species proposed using two HRM amplicons was possible. Initially, an amplicon with 82 bp was used and the species *L. eximia*, *L. cuprina*, *C. putoria*, and *Co. macellaria* were distinguished, but *C. megacephala* and *C. albiceps* presented the same melting curves, as shown in Fig. 1. Complementarily, a second amplicon with 124 bp was used and it easily distinguished *C. megacephala*, *Co. macellaria, C. albiceps*, and *C. putoria*. The species *L. cuprina* and *L. eximia* could not be distinguished due to the similarity of their melting curves (Fig. 2). The second amplicon 124 bp allowed identifying the species that 82 bp amplicon could not differentiate (Fig. 3).

The distinction of *L. cuprina* and *L. eximia* species by the 82 bp HRM amplicon was advantageous since most samples of *L. eximia* could not be amplified and sequenced using universal COI primers (658 bp fragment), only when we used a short fragment (385 bp). This fact led us to believe that *L. eximia* samples would be degraded (5 of 6 samples were in a dry state). Table 2 shows that the only *L. eximia* sample that could be identified by DNA sequencing was the one preserved in alcohol, which suggests DNA was probably degraded in those samples. DNA degradation could also explain why amplifying most of the old samples with the universal DNA barcoding primers was not possible (*Hajibabaei et al., 2006*). It also occured for old samples of *C. albiceps* and *C. putoria*. The amplification of large fragments does not occur when the DNA is broken in smaller fragments, but a shorter marker can be used for identification (*Boyer et al., 2012*).

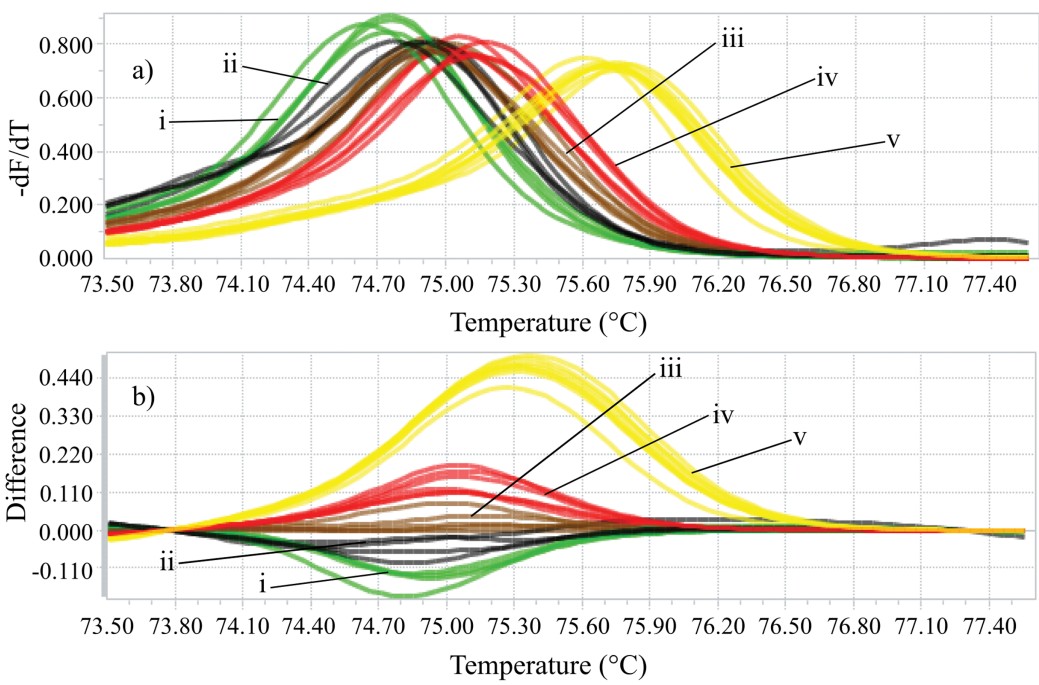

**Figure 2 High-resolution melting analysis using COI primers (124 bp amplicon) for forensic species.** (A) Normalized melting curves. (B) Difference plot curves using *C. albiceps* as reference genotype. Yellow: *C. megacephala*; Red: *Co. macellaria*; Brown: *C. albiceps*; Black: *L. cuprina* and *L. eximia*; Green: *C. putoria*. (i) *C. putoria*; (ii) *L. cuprina* and *L. eximia*; (iii) *C. albiceps*; (iv) *Co. macellaria*; (v) *C. mega-cephala*.

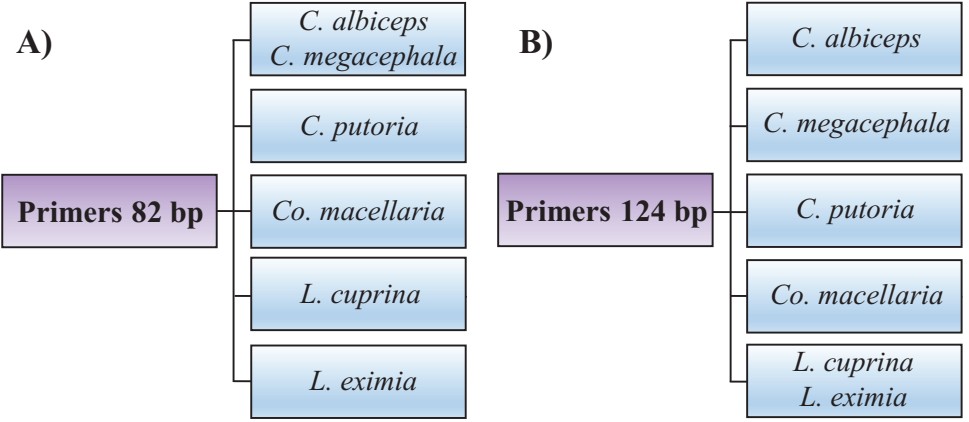

**Figure 3 Flowchart showing species identification of blowflies using each of the HRM primers proposed, (A) 82 bp amplicon and (B) 124 bp amplicon.**

Our results confirmed the efficiency of alcohol 70 °G.L in the preservation of samples. The preservation in alcohol allowed the generation of DNA barcodes using the largest universal primers for 100% of the samples, even for old samples. This confirms that the use of this solvent can be an important ally in long-term conservation of flies, improving the sequence recovery rates (*Elías-Gutiérrez et al., 2018*). Similar results were also reported for other insects (*Stein et al., 2013*).

**Table 2 Percentage of samples with results using DNA sequencing and High-Resolution Melting when preservation condition and time since collection were analyzed.**

| Condition | Number of samples | 650 bp fragment (%) | 300 bp fragment (%) | HRM 82 bp (%) | HRM 124 bp (%) |
|---|---|---|---|---|---|
| Dry | 13 | 0 | 100 | 92 | 46 |
| Alcohol | 25 | 100 | * | 92 | 80 |
| Fresh | 5 | 100 | * | 100 | 40 |
| Total | 43 | 68 | 30 | 91 | 65 |
| Ancient | 38 | 66 | * | 92 | 68 |
| Fresh | 5 | 100 | * | 100 | 40 |
| Total | 43 | 70 | 30 | 91 | 65 |

**Note:**
* Samples sequenced only using 658 bp amplicon.

HRM was superior to DNA sequencing for dried (and possibly degraded) samples when the storage condition was considered, since each sample was amplified using the universal primers. The result was superior using 82 bp and similar using 124 bp amplicon in all of these samples, including the fresh samples and the samples preserved in alcohol. Thus, dry or degraded samples could be assayed with confidence by HRM.

The 82 bp amplicon showed to be effective for the molecular analysis of old specimens DNA with 92% of sample identification when the storage time was considered, while 124 bp amplicon identified only 68%. This 82 bp amplicon presented 92% of sample identification against 66% when compared with DNA sequencing. We only used specimens with more than a decade and degraded DNA to show HRM analysis is sufficient to distinguish species in difficult conditions.

Although DNA sequencing has been considered the best technique for species identification, it could not be efficient at identifying degraded samples amplifying large fragments. Moreover, this procedure is laborious and expensive. The HRM technique overcomes these problems since it can amplify and distinguish even species with degraded DNA (*Boyer et al., 2012*). This is a closed tube technique, reducing the contamination risks without using toxic reagents. In addition, the working time is short; sample analysis, detection of DNA polymorphisms, and distinctions between the melting curves can last about 2 h. Furthermore, HRM technique is cheaper than sequencing. Thus, adopting the HRM technique in laboratory routine and genetic studies is possible.

## CONCLUSIONS

Our results support that the HRM analysis using our COI primer set is a powerful tool and sensitive technique for the identification and distinction of occurring Calliphoridae species in Brazil. The two amplicons designed can be reliably used to determine species identity, especially when morphological identification is not possible. Moreover, even ancient specimens collected and preserved dried for more than ten years, with possible degraded DNA, could be identified, unlike what occurs when using the DNA sequencing technique, which failed for those samples. New HRM assays should be performed in other blowfly forensic groups to facilitate the routine identification of species.

## ACKNOWLEDGEMENTS

We thank Dr. Patrícia Jacqueline Thyssen for providing us with the samples used in this study, Dr. Márcia Flores da Silva Ferreira for the authorization to use the real time PCR equipment. We also thank the Graduate Program in Biotechnology of the Federal University of Espirito Santo, and group research, Geotechnology Applied to Global Environment (GAGEN).

### Funding

This study was supported by the Coordenação de Aperfeiçoamento de Pessoal de Nível Superior - Brazil (CAPES) – Finance Code 001, for the scholarship awarded to the Pablo Viana Oliveira. There was no additional external funding received for this study. The funders had no role in study design, data collection and analysis, decision to publish, or preparation of the manuscript.

### Grant Disclosures

The following grant information was disclosed by the authors:
Coordenação de Aperfeiçoamento de Pessoal de Nível Superior - Brazil (CAPES): 001.

### Competing Interests

The authors declare that they have no competing interests.

### Author Contributions

- Pablo Viana Oliveira conceived and designed the experiments, performed the experiments, analyzed the data, prepared figures and/or tables, authored or reviewed drafts of the paper, and approved the final draft.
- Francine Alves Nogueira de Almeida performed the experiments, authored or reviewed drafts of the paper, and approved the final draft.
- Magda Delorence Lugon performed the experiments, authored or reviewed drafts of the paper, and approved the final draft.
- Karolinni Bianchi Britto performed the experiments, authored or reviewed drafts of the paper, and approved the final draft.
- Janyra Oliveira-Costa conceived and designed the experiments, authored or reviewed drafts of the paper, and approved the final draft.
- Alexandre Rosa Santos analyzed the data, prepared figures and/or tables, authored or reviewed drafts of the paper, and approved the final draft.
- Greiciane Gaburro Paneto conceived and designed the experiments, analyzed the data, authored or reviewed drafts of the paper, and approved the final draft.

### Data Availability

The raw data are available as Supplemental Files and include Calliphoridae species samples and the COI sequences downloaded from BOLD Systems and GenBank (Table S2).

## Supplemental Information

Supplemental information for this article can be found online at http://dx.doi.org/10.7717/peerj.9680#supplemental-information.

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
