# Peer review of "Using high-resolution melting to identify Calliphoridae (blowflies) species from Brazil"

_PeerJ, doi:10.7717/peerj.9680_

## Round 0.1 · original submission · Major Revisions

Dear Dr. Oliveira and colleagues:

Thanks for submitting your manuscript to PeerJ. I have now received three independent reviews of your work, and as you will see, the reviewers raised some concerns about the research. Despite this, these reviewers are optimistic about your work and the potential impact it will have on research studying blowfly forensics and systematics. Thus, I encourage you to revise your manuscript, accordingly, taking into account all of the concerns raised by both reviewers.

While the concerns of the reviewers are relatively minor, this is a major revision to ensure that the original reviewers have a chance to evaluate your responses to their concerns. There are not too many suggestions; thus, it should not take much effort to address these concerns to greatly improve your manuscript.

You must provide evidence for duplicated experiments and for triplicates in each HRM run (either technical and or biological). Please also present the confidence values for the HRM profiles. Please also be more descriptive in explaining your methodologies.

I look forward to seeing your revision, and thanks again for submitting your work to PeerJ.

Good luck with your revision,

-joe

·

Basic reporting

The manuscript is written in clear proffessional language. Introduction show advanteges and limitations of existing methods of species identification important for forensic entomology. Structure of the manuscript conforms PeerJ standards. Figures are high quality and relevant and well described. Provided References are well selected and up to date.

Experimental design

Performed research are in scope of the Journal. Research aim - delivering simple and robust method for identification of forensically inportant Calliphorode species in Brasil - is well defined. To reach this aim is necessary to fill the gap - design approptiate primers. The Authors successfully done it.

Description of methods, however is insufficient.
Line 70 and 71. „Mitochondrial COI barcode sequences of those species were downloaded from BOLD systems“. What were the criteria for selecting the sequence to be analyzed?.
For example, in Supplementary Table 2 authors chosed 5 sequences of Chrysomya megacephala for analysis. For this species in BOLD there are 833 public records with sequences, 47 of its from Brazil, but the Authors did not selected any sequence of Chrysomya megacephala from Brazil. Why?

Line 73. The Authors selected two regions containing polymorphic variations for species distinction. It would be uselul include sequences alignment in supplementary material. The Authors do not indicate what the differences were in the amplicon sequences.

Line 75-76. uMelt-DNA Melting software version (available at https://www.dna.utah.edu/umelt/um.php) - link outdated

Validity of the findings

The Authors provide well supported data. Forensic entomology will benefits from publication of this manuscript.

Reviewer 2 ·

Basic reporting

The article is very well written and the literature cited is sufficient. Figures and tables are good.

Experimental design

The article offers an important methodology to be used to differentiate insect species of forensic interest and

Validity of the findings

The results are well-founded and the text presents a good discussion just like the conclusion.

Additional comments

Calliphorids are very similar and therefore the use of methodologies that can differentiate the species is always very important.

I missed the article to include quotes from works that use scanning electron microscopy to differentiate species in both immature and adult stages so it is also an important tool.

Reviewer 3 ·

Basic reporting

This is an interesting manuscript using Barcoding coupled with HRM analysis for forensic analysis and identification of insects. The manuscript is well written and the authors have used English proofreading. The introduction is describing the problem and the methods. The literature is adequate and the figures and tables describe clearly the results and the method

Experimental design

The experiments are appropriately designed. The question is well defined and the methodology to answer is appropriate. what is not clear it is that there are no replicates for the HRM and no triplicates indicated for each sample which should be. Each sample should run in triplicate, (biological or/and technical triplicates)

Validity of the findings

The results are robust except for the replicates and triplicates that the authors should provide, the authors should also provide the confidence value for each sample run in the HRM

Additional comments

The authors state in line 15 ancient DNA but this is only DNA from stored dried samples, the authors should change the word. In lines 56-57 and in the discussion the authors refer to the pros of the HRM but they should also refer the cons and the constrains of the method. Lines 129-130 need clarification

---

## Round 0.2 · accepted · Accept

Dear Dr. Oliveira and colleagues:

Thanks for revising your manuscript based on the concerns raised by the reviewers. I now believe that your manuscript is suitable for publication. Congratulations! I look forward to seeing this work in print, and I anticipate it being an important resource for groups studying blowfly forensics and systematics. Thanks again for choosing PeerJ to publish such important work.

Best,

-joe